# Health, environmental, and animal rights motives for vegetarian eating

**Christopher J. Hopwood**◉*, Wiebke Bleidorn, Ted Schwaba, Sophia Chen

University of California, Davis, CA, United States America

* chopwoodmsu@gmail.com

## Abstract

Health, the environment, and animal rights represent the three main reasons people cite for vegetarian diet in Western societies. However, it has not been shown that these motives can be distinguished empirically, and little is known about what kind of people are likely to be compelled by these different motives. This study had three goals. First, we aimed to use construct validation to test whether develop health, environmental, and animal rights motives for a vegetarian diet could be distinguished. Second, we evaluated whether these motivations were associated with different demographic, behavioral, and personality profiles in three diverse samples. Third, we examined whether peoples' motivations were related to responses to vegetarian advocacy materials. We created the Vegetarian Eating Motives Inventory, a 15-item measure whose structure was invariant across three samples (N = 1006, 1004, 5478) and two languages (English and Dutch). Using this measure, we found that health was the most common motive for non-vegetarians to consider vegetarian diets and it had the broadest array of correlates, which primarily involved communal and agentic values. Correlates of environmental and animal rights motives were limited, but these motives were strong and specific predictors of advocacy materials in a fourth sample (N = 739). These results provide researchers with a useful tool for identifying vegetarian motives among both vegetarian and non-vegetarian respondents, offer useful insights into the nomological net of vegetarian motivations, and provide advocates with guidance about how to best target campaigns promoting a vegetarian diet.

**Data Availability Statement:** Pre-registration, methods, measures, scripts, and supplemental results for samples 1-3 as well as data for samples 1 and 2 are available at https://osf.io/52v6z/. Data for sample 3 cannot be shared publicly because it

Eating is an important day to day behavior at the interface of individual differences, social dynamics, economics, health, and ethics. Vegetarianism has emerged as a significant dietary movement in Western cultures [1–3]. The benefits of vegetarian diets include improved individual health [4–8], a more sustainable environment [4,9–11], and a more humane approach to inter-species relationships [12–19].

Health, environment, and animal rights also appear to represent the primary non-religious motives for a plant-based diet [1,20–24]. However, thus far there is very little evidence that these motives can be distinguished empirically, and no existing measures of eating behavior is available to measure health, environment, and animal rights as distinct motives for vegetarian diet. One consequence of this gap in the literature is that relatively little is known about the

is not owned by the authors. It can be requested at https://www.lissdata.nl. Preregistration, materials, and data for sample 4 are available at https://osf.io/9wre4/.

**Funding:** Funding was provided to Christopher J. Hopwood and Wiebke Bleidorn by Animal Charity Evaluators (https://animalcharityevaluators.org). The funding agency advised on study design issues prior to data collection; all decisions about study design were determined by the authors.

**Competing interests:** The authors have declared that no competing interests exist.

psychological implications of these different reasons for a vegetarian diet. Initial research suggests that extraverted and sociable individuals tend to be more motivated by health [25,26] whereas factors such as agreeableness, openness, altruism, and empathy may be more related to ethical motivations [27,28]. However, findings are often inconsistent, and a wide range of potentially important correlates have not been examined. Understanding these motives is important for advancing knowledge about this increasingly important behavior, and it may also have practical value in the area of advocacy.

Advocates for plant-based diets typically focus on at least one of these three motives when trying to convince people to adopt a plant-based diet or join a vegan organization [20,29–31]. Advocacy campaigns may be more effective to the degree that they target the specific motives of different groups and individuals [30,32] because people are more likely to respond to messages that target their personal needs and interests [33]. Moreover, focusing on issues that do not resonate with individuals' motives may negatively impact animal advocacy, such as when the exposure to animal rights advocacy creates an unpleasant emotional reaction [34] that worsens opinions of vegetarians and animal advocacy [31]. Thus, it is in the interest of advocacy groups to better understand the kinds of people who are more or less likely to respond to activism that emphasizes health, the environment, or animal rights. From an advocacy perspective, it is particularly important to understand the motives to which non-vegetarians are most sympathetic, given that these are the individuals that are targeted by advocacy campaigns.

The goals of this research were to 1) evaluate the structure of common motives for a vegetarian diet, 2) to use that measure to develop behavioral and psychological profiles of people who would be most likely to adopt a plant-based diet for different reasons, and 3) examine whether this profile predicts responses to advocacy materials.

## Motives for a plant-based diet

Many instruments have been developed to assess diet-related motives. Early work tended to focus on specific motives of interest for a particular research topic. For example, Jackson, Cooper, Mintz, and Albino [35] created a scale focused on eating motives in the context of substance abuse, which included four dimensions: coping, social motives, compliance, and pleasure. While this instrument outlines a useful model of psychological eating motives, it is less suitable for research on vegetarian diet because any of these four motives could lead a person to eat either vegetarian or non-vegetarian food, depending on other considerations.

Several instruments tap eating motives that are more likely to distinguish vegetarian from non-vegetarian eaters. The *Food Choice Questionnaire* (FCQ; [36]) focuses on nine motives: convenience, price, health, sensory appeal, weight control, natural content, mood, familiarity, and ethical concerns. Renner, Sproesser, Strohbach, & Schupp [37] developed *The Eating Motivations Survey* (TEMS), a broad, multidimensional measure of 15 different motives including liking, habits, hunger, health, convenience, pleasure, tradition, nature, sociability, price, visual appeal, weight control, affect regulation, social norms, and social image. These multiscale measures provide a general taxonomy of individual motivations in food choice, but they do not distinguish the three core motives most central to vegetarian diets, and they include a variety of motives that are less relevant for plant-based diets such as mood or affect regulation.

Other measures have focused more specifically on ideological or ethical factors potentially more relevant to vegetarianism. Lindeman and Stark [38] created a measure with scales designed to distinguish ideological reasons, weight control, health, and pleasure. In a similar project, Arbit, Ruby, and Rozin [39] crafted the *Meaning in Food Life Questionnaire* (MFLQ), which has three dimensions, social, sacred (i.e., religious), and aesthetic, that are not relevant

to our study, and two that are: moral (which could include animal rights and environmental motives) and health. Lindeman and Väänänen [40] set out to enhance the FCQ by developing four scales focused on ethical dimensions, including animal welfare, the environment, politics (e.g., human rights related to food production), and religion. However, in their study, the animal welfare and environment scales were so highly correlated that they collapsed into a single factor. Measures focused on ethical motivations for food choice begin to capture variation in motives that might be specific to vegetarian diets, but they tend to collapse different ethical concerns relevant to vegetarian diet into a single factor and don't always include health. Indeed, distinguishing various ethical factors may be difficult in practice [21,41,42], as results from these studies also show that even when items are identified to distinguish moral from health-related motives, it is challenging to distinguish these motives in terms of external correlates. An important exception is the *Dietarian Identity Questionnaire* [2], which has scales designed to measure a range of dimensions that link dietary behavior to identity, including the emphasis an eater places on prosocial as opposed to moral concerns when making food choices. This framework has considerable promise for identifying the mechanisms underlying these different motivations for vegetarian diets (e.g., Rosenfeld, 2019 [43]), but it does not provide scales to directly measure health, environmental, and animal rights motives for a vegetarian diet.

Thus, the first step in our research was to use a construct validation strategy to test whether the three main reasons people might have adopted or be compelled to adopt a plant-based diet —health, animal rights, and the environment—can be distinguished empirically. Given ambiguities in the literature, we focused specifically on differentiating environmental and animal rights factors.

## Identifying characteristic profiles of people with different vegetarian motives

Variables related to plant-based eating in general include younger age [44,45], being female [1,44,46–49], living in urban areas [50–54], and having liberal values [45,46,49,52,55–59]. Thus, vegetarians can be reliably characterized, to some degree, in broad strokes.

Yet, different vegetarians can arrive at a plant-based diet for very different reasons. How are people who are primarily motivated by their personal health different from people who are primarily motivated by their concerns about the environment or their compassion for animals? The second goal of this project is to distinguish people who are most likely to pursue plant-based diets for reasons related to their personal health, the environment, or animal rights. Distinct profiles of people with these different motives could help advocacy campaigns reliably identify individuals and groups who are most likely to respond to their message.

Given the limited evidence regarding correlates of different motivations and the fact that there is a wide range of plausible correlates, our overall approach was to include an extensive array of possible attributes with plausible links to vegetarian motives and to use multiple samples and increasingly strict statistical tests to hone in on replicable associations. We included attributes related to demographic characteristics, personality traits, values, hobbies, religious background and behavior, habits, entertainment preferences, and patterns of social media use. We then 1) identified potential correlates in an American undergraduate convenience sample, 2) identified which associations replicate in an American community convenience sample, and 3) tested preregistered hypotheses, based on these replicated associations, about which variables would replicate in a large representative Dutch sample. We reasoned that any associations observed consistently across all three of these samples would be sufficiently robust to be useful for informing research on motives for plant-based eating and for guiding advocacy efforts.

## Vegetarian motives and responsiveness to advocacy materials

The motivational complexity of vegetarian behavior implies that advocacy will generally be most effective if it targets the specific motives of its audience. This is presumably why advocacy groups tend to campaign on one of the three main reasons to adopt plant-based diets—health, the environment, and animal rights. But is it true that people with different levels of health, environmental, and animal rights motives will be differentially sensitive to advocacy materials that target their primary motives? The third goal of this project was to use the measure we developed to determine whether individual differences in motives for vegetarian eating predict responsiveness to advocacy materials that focus on health, the environment, or animal rights.

## Method

This study was approved by the UC Davis IRB #1145613–1 and #1372555–2.

### Sample 1

Our first sample consisted of 1006 undergraduates attending a public university in the United States who participated in exchange for course credit. The mean age of these students was 19.80 (SD = 3.33); 822 (81.7%) were female, 180 (17.9%) male, and 4 (0.4%) nonbinary. Racial composition was 485 Asian (48.2%), 22 black (2.2%), 47 Latin American (4.7%), 27 Native American (2.7%), 328 white (32.6%), 94 multiracial (9.3%), and 3 other (0.3%); 252 (25.0%) reported Hispanic ethnicity. Eleven participants self-identified as vegan and 44 as vegetarian.

### Sample 2

Our second sample consisted of 1004 Amazon MTurk Workers who completed a survey for financial compensation (prorated at $10/hour). The average age in this sample was 36.46 (SD = 10.99); 471 (46.91%) were female, 532 (53.00%) were male, and 1 (0.1%) was nonbinary. Ethnic/racial composition was 63 (6.7%) Asian, 113 (11.3%) black, 111 (11.1%) Hispanic, 10 (1.0%) Native American, 780 (77.7%) white, 32 (3.2%) multiracial, and 6 (0.6%) other. Participants in this sample were not restricted based on geography. Seventeen participants self-identified as vegan and 25 as vegetarian.

### Sample 3

Our third sample included 5478 Dutch participants drawn from the Longitudinal Internet Studies of the Social Sciences (LISS). The mean age in this sample was 51.34 (SD = 18.31); 3,106 (54.0%) were female and 2,642 (46.0%) were male. Sixty-nine participants self-identified as vegan; vegetarian identity was not assessed in the LISS sample.

### Sample 4

Our fourth sample consisted of 739 undergraduate participants (mean age = 20.01, SD = 3,60; 615 women (83.0%); 186 Hispanic (25.0%) ethnicity; 178 white (24.0%), 10 black (1.4%), 363 Asian (49.0%), 4 Pacific Islander (0.5%), 84 multiracial (11.4%), and 95 other race (12.9%). Eight people reported vegan diet and 27 reported vegetarian diet.

The only exclusion criterion across samples was being 18 years or older. Participants were not excluded based on dietary habits or preferences.

## Instrument development strategies

Based on an initial literature review, we generated 26 items designed to assess health, environmental, and animal rights motives for a plant-based diet. We administered these items to participants in Sample 1 and conducted a series of item-level factor analyses to identify a reduced set of items that loaded onto the three factors with strong pattern coefficients and minimal cross-loadings. We then administered and examined this reduced set of items in Sample 2. We examined the fit of the measurement model within both samples and measurement invariance across both samples using confirmatory factor analysis (CFA). Items, instructions, and response scales for the final version of the instrument, which we called the *Vegetarian Eating Motives Inventory* (VEMI), are given in Table 1.

We translated the VEMI items into Dutch in order to administer it to Sample 3. We first asked a native Dutch speaker who also speaks English to translate the items. We then asked a native English speaker who also speaks Dutch to back translate them. The research team confirmed that the content was retained for all items through this process. We evaluated the fit of the measurement model and measurement invariance using CFA. Items, instructions, and response scales for the Dutch version of the VEMI are available at https://osf.io/wyfgb/.

## Validating measures

We sought to measure a wide range of variables that could plausibly distinguish motives for a plant-based diet. Our main constraint was the measures that already existed in the LISS data (i.e., Sample 3) to whom we would administer the VEMI but whose data collection was otherwise already planned. Overall, we assessed 260 characteristics (https://osf.io/y8nd5/). These characteristics included demographic features, personality traits, terminal and instrumental values, religious beliefs and behaviors, involvement in various organizations and volunteer activities, employment/income, hobbies/interests, online behavior and preferences, social behavior, and habits.

**Table 1.  Vegetarian Motives Inventory (VEMI).**  Please rate the importance of each of the following reasons for you to eat less meat or animal products. Please rate these items even if you don't intend to change your diet.

| Scale: | | | | | | |
|---|---|---|---|---|---|---|
| **ss1** | **2** | **3** | **4** | **5** | **6** | **7** |
| **Not important** | | **Moderately important** | | | **Very important** | |

| |
|---|
| 1. I want to be healthy (H) |
| 2. Plant-based diets are better for the environment (E) |
| 3. Animals do not have to suffer (A) |
| 4. Animals' rights are respected (A) |
| 5. I want to live a long time (H) |
| 6. Plant-based diets are more sustainable (E) |
| 7. I care about my body (H) |
| 8. Eating meat is bad for the planet (E) |
| 9. Animal rights are important to me (A) |
| 10. Plant-based diets are environmentally-friendly (E) |
| 11. It does not seem right to exploit animals (A) |
| 12. Plants have less of an impact on the environment than animal products (E) |
| 13. I am concerned about animal rights (A) |
| 14. My health is important to me (H) |
| 15. I don't want animals to suffer (A) |

## Strategy for identifying correlates of vegetarian motives

Our general approach to identifying specific motive-outcome associations in order to pre-register hypotheses for Sample 3 was to estimate a series of multiple latent regressions using the R package *lavaan* [60] in Sample 1 that we then attempted to replicate in Sample 2. First, we estimated six different models separately: one in which all associations between outcome and the three latent eating motives variables were constrained to be equal (model All Equal), one in which all motive-outcome associations were constrained to zero (model All Zero), three in which one motive-outcome association was estimated freely but the other two motives were constrained to have equal associations with the outcome (models Animal Free, Environment Free, and Health Free), and one in which all motive-outcome associations were estimated freely (model All Free).

We then conducted a series of nested $\chi^2$ model comparison tests for each motive-outcome association to identify which of these six models best fit the data. We first compared the fit of model All Equal to model All Zero. If model All Zero did not fit significantly worse ($p < .05$), we selected model All Zero as the best fit and concluded that no eating motives were significantly associated with the outcome variable. If model All Zero fit worse than model All Free, we compared the fit of model All Equal to whichever of Animal Free, Environment Free, and Health Free fit best to the data (as these models have equal degrees of freedom, they were not nested; the best-fitting model was identified as the one with the lowest $\chi^2$ and BIC values). If none of these models fit significantly better than model All Equal, we selected model All Equal as the best fit and concluded that the three eating motives were not differentially associated with the outcome variable. However, if Animal Free, Environment Free, or Health Free models fit significantly better to the data than model All Equal, we compared the fit of that model versus the fit of model All Free. If model All Free fit significantly better, we concluded that eating motives were differentially associated with the outcome variable. If All Free did not fit significantly better, and Animal Free, Environment Free, or Health Free was the best fitting model, we concluded that one specific motive was differentially associated with the outcome variable. The R code used to perform these analyses is available at https://osf.io/49shv/.

Next, we examined whether any patterns of non-zero motive-outcome associations replicated in the MTurk sample. To do this, we estimated two multiple-groups models in lavaan. In the first model (model Replication), motive-outcome associations from the best-fitting model identified in Sample 1 (model All Free, Animal Free, Health Free, Environment Free, or All Equal) were imposed to be equal across both samples. In the second model (model Nonreplication), motive-outcome associations in Sample 1 were constrained to the best-fitting model, while motive-outcome associations in Sample 2 were freely estimated. We compared the fits of these two nested models using a $\chi^2$ model comparison test. If model Nonreplication fit the data significantly better ($p < .05$), we concluded that the pattern of associations did not replicate across samples. Otherwise, we concluded that the pattern of associations in Sample 1 replicated in Sample 2.

Although the aforementioned steps described our primary procedure, it had two important limitations. First, inspection of the path coefficients revealed instances when very similar effect sizes across samples were classified as non-replications. Second, because these analyses used multiple regressions, they were also prone to suppression effects. We therefore contextualized these initial results with two additional rules. First, to restrict our interpretations to meaningful effects, we examined whether any moderate-or-stronger associations between specific eating motives and outcomes replicated across samples. To do this, we first identified all outcomes for which one or more motive-outcome associations was stronger than Beta weights = |.15| in both samples. We only retained variables with an effect of |.15| or larger. Second, to avoid

interpreting effects that were only present due to statistical suppression, we examined the bivariate correlations for each replicated motive-criterion association in the first two samples and discarded the cases in which the regression coefficient and bivariate correlation were of opposite signs or in which the bivariate correlation was < |.15|.

### Vegetarian motives and responsivity to advocacy flyers

We conducted a pre-registered validation study to test the sensitivity of the VEMI scales to attitudes about advocacy flyers specifically appealing to health, environmental, and animal rights motives for a plant-based diet (see https://www.vegansociety.com). Participants answered six questions about each flyer (e.g., this flyer made me want to be vegan) on a scale from 1–7. Internal consistencies were above .90 for these sets of questions for all three flyers, and an item-level factor analysis provided strong support for a single factor. We predicted that scores on the VEMI motives scales would be specifically associated with positive attitudes about the flyer targeting that motive (e.g., health motives would be related to positive attitudes about the health flyer) as indexed by both significant bivariate correlations and significant Beta weights in regression models in which all three VEMI scales are regressed upon the attitude scales, one at a time.

## Results

Pre-registration, methods, measures, scripts, and supplemental results for samples 1–3 as well as data for samples 1 and 2 are available at https://osf.io/52v6z/. Data for sample 3 can be requested at https://www.lissdata.nl. Preregistration, materials, and data for sample 4 are available at https://osf.io/9wre4/.

### Developing the Vegetarian Motives Inventory (VEMI)

Fifteen items were chosen from the original pool of 26 (Table 1) based on exploratory factor analyses in Sample 1. The model fit the data well and was invariant across all three samples (Table 2). It was also invariant across men and women and across white vs. non-white participants in samples 1 and 2 (Table 2). Cronbach's alpha estimates of internal consistency across the three samples, respectively, were .88, .91, and .89 for the health scale, .90, .94, and .92 for the environment scale, and .93, .96, and .94 for the animal rights scale. Latent correlations between these scales in the three samples, respectively, were .33, .40, and .43 between health and environment, .27, .35, and .49 between health and animal rights, and .57, .70, and .59 between environment and animal rights.

VEMI scale means across our first three samples are given in Table 3. In general, people tended to respond above the raw midpoint of 4, indicating that health, the environment, and animal rights are all considered to be generally compelling reasons to adopt a plant-based diet. This was particularly the case for the health scale, for which the mean approached 6 (out of 7) in all three samples. As a validity check, we also asked participants in Samples 1 and 2 to rank the main reason they would choose to adopt a plant-based diet. Of the 1826 participants who responded to this question, the standardized means for corresponding VEMI scales were consistently ranked as the most important reason (e.g. people who rated Health highest on the VEMI scale also tended to rank Health as their main reason to adopt a plant-based diet). Again, these results showed that health is the most common reason among this primarily non-vegetarian sample to consider eating less meat, as 75% of respondents ranked this motive first. Finally, large effects distinguished the 97 vegans across all three samples from non-vegan respondents for the health (d = .51), environment (d = 1.29), and animal rights (d = .97) scales (all $p < .001$; Table 4).

**Table 2. CFA model fit for the VEMI in three samples.**

|  | df | $\chi^2$ | CFI | RMSEA |
|---|---|---|---|---|
| Undergraduate sample 1 | 87 | 359.97 | .975 | .056 |
| MTurk sample 2 | 87 | 462.52 | .975 | .065 |
| LISS sample 3 | 87 | 3813.69 | .948 | .086 |
| Invariance tests across three samples |  |  |  |  |
| Configural | 261 | 4636.19 | .955 | .080 |
| Constrain factor loadings | 285 | 4804.38 | .954 | .078 |
| Constrain intercepts | 309 | 5978.15 | .942 | .084 |
| Constrain latent means | 315 | 6306.12 | .939 | .086 |
| Invariance tests across males and females in three samples (N = 7,753) |  |  |  |  |
| Configural | 174 | 3884.0 | .962 | .074 |
| Constrain factor loadings | 186 | 3906.7 | .962 | .072 |
| Constrain intercepts | 198 | 3969.1 | .961 | .070 |
| Constrain latent means | 201 | 4166.3 | .959 | .071 |
| Invariance tests across white and non-white participants in samples 1 and 2 (N = 2,010) |  |  |  |  |
| Configural | 174 | 767.38 | .977 | .058 |
| Constrain factor loadings | 186 | 784.62 | .977 | .057 |
| Constrain intercepts | 198 | 813.65 | .976 | .056 |
| Constrain latent means | 201 | 822.39 | .976 | .055 |

df = degrees of freedom. CFI = Confirmatory Fit Index. RMSEA = Root Mean Square Error of Approximation. We used model comparison tests based on fit indices to examine measurement invariance. We established measurement invariance (all ΔCFI < .01 or all ΔRMSEA < .01; cf. Cheung & Rensvold, 2002). That is, we were able to constrain configuration, factor loadings, intercepts, and latent means across groups without a significant decrease in model fit.

## Identifying correlates of plant-based eating motives

Based on an initial examination of criterion variable distributions, the following variables were log-transformed in order to normalize distributions: gross monthly income, all values, weekly hours volunteering, weekly hours spent watching sports, weekly hours watching tv, weekly hours listening to the radio, number of books read in the last 30 days, frequency of social media use, and hours per week spent online. We also log-transformed these variables in Samples 2 and 3. We excluded 49 binary variables with insufficient variance in either Samples 1 or 2 (i.e., less than 50 participants responding either "no" or "yes") and 4 continuous variables with no variance in Samples 1 or 2. We did not consider any other variables in the LISS sample that were not also assessed in Samples 1 and 2. Given these exclusions, we examined associations between VEMI scales and 207 remaining criterion variables.

We first computed bivariate correlations between VEMI scales and the 207 criterion variables. The 56 criterion variables with replicated associations (p < .01) across all three samples are presented in Table 5. Among those, most variables correlated with all three motives, with health motives uniquely, or with both health and animal rights motives.

**Table 3. Means and standard deviations for VEMI scales in three samples.**

|  | Health | Environment | Animal Rights |
|---|---|---|---|
| Undergraduate sample 1 | 5.93 (1.07) | 4.38 (1.39) | 4.64 (1.46) |
| Mturk sample 2 | 5.71 (1.30) | 4.25 (1.73) | 4.35 (1.79) |
| LISS sample 3 | 5.88 (1.09) | 4.80 (1.20) | 5.27 (1.47) |

**Table 4. Standardized means for three VEMI scales among people who ranked health, environment, and animal rights as most important factor in considering a vegetarian diet.**

| Ranking | N | Health Scale | Environment Scale | Animal Rights Scale |
|---|---|---|---|---|
| Health | 1366 | .08 (.93) | -.18 (.98) | -.21 (.98) |
| Environment | 247 | -.25 (1.15) | .54 (.88) | .28 (.83) |
| Animal Rights | 213 | -.31 (1.26) | .34 (1.04) | .96 (.75) |

As described above, our primary analytic approach used a regression-based strategy in a latent-variable framework to test preregistered predictions in sample 3 based on results from samples 1 and 2. Among the 207 criterion variables, we identified 33 that were significantly associated with at least one VEMI scale in both of the first two samples. Table 6 shows the results of the best-fitting models for those criterion variables. We based predictions for Sample 3 based on two criteria from analyses of data from samples 1 and 2.: a positive standardized path coefficient of $|.15|$ or larger and a bivariate correlation of $|.15|$ or larger. Based on these results, we predicted that a) valuing peace would be related to all three motives (in this case we relaxed our rule somewhat; although the regression coefficient for animal motives was .14 in the second sample, bivariate correlations were virtually identical across variables), b) agreeable personality, valuing truth, responsibility, hard work, forgivingness, courage, helpfulness, lovingness, self-control, independence, instrumental happiness, intellect, family security, freedom, self-respect, terminal happiness, wisdom, national security, salvation, friendship, accomplishment, harmony, comfort, and mature love would have specific associations with health motives, c) being involved with an environmental organization would have a specific association with environmental motives, and d) caring for plants or animals would have a specific association with animal rights motives. Seven variables with standardized regression coefficients above our threshold in both samples did not have bivariate correlations < $|.15|$ and thus we predicted they would not be related to any plant-based eating motives in the LISS data. The preregistration document for Sample 3 based on these findings can be found at https://osf.io/rk4en/. We mistakenly made predictions about three variables based on results in sample 1 and 2 that were not available in LISS—being vegetarian, eating meat, and being involved in an animal organization.

Associations that met the replication criteria described in the preceding paragraph are given in Table 7. Overall, 16 variables were related specifically and positively to health motives, including the personality trait agreeableness and a number of different values. The only variable that was related specifically to environmental motives was participation in an environmental organization. No variables were related specifically to animal rights motives.

## Vegetarian motives and responsivity to advocacy flyers

Participants from Sample 4 completed the VEMI and answered questions about advocacy flyers targeting health, environment, and animal rights motives created by The Vegan Society. We used these data to test pre-registered hypotheses about the specificity of correlations between the VEMI scales and attitudes about flyers targeting health, environment, and animal rights motives (https://osf.io/9wre4/). Table 8 shows that all bivariate correlations between motives and responses to flyers were statistically significant ($p < .05$). As predicted, the strongest correlate of the environment flyer was the VEMI environment scale and the strongest correlate of the animal rights flyer was the VEMI animal rights scale. Inconsistent with our hypotheses, both the environment and animal rights scales were also more strongly correlated with responses to the health flyer, suggesting that people who are motivated by health are not particularly impacted by vegetarian advocacy, in general.

**Table 5. Bivariate correlations between plant-based motives and criterion variables for which at least one motive correlated significantly (p < .01) across samples 1, 2, and 3.**

| Sample | Health | | | Environment | | | Animal Rights | | |
|---|---|---|---|---|---|---|---|---|---|
| | **1** | **2** | **3** | **1** | **2** | **3** | **1** | **2** | **3** |
| Male | **-.08** | **-.08** | **-.11** | **-.16** | -.04 | **-.15** | **-.28** | -.07 | **-.18** |
| Extraversion | **.17** | **.16** | **.14** | .06 | **.16** | .03 | -.01 | **.11** | .04 |
| Agreeableness | **.26** | **.32** | **.21** | **.15** | **.20** | **.19** | **.21** | **.17** | **.23** |
| Conscientiousness | **.15** | **.23** | **.20** | .06 | **.09** | .07 | **.13** | .07 | **.09** |
| Neuroticism | **.10** | **.11** | **.09** | -.05 | -.02 | -.03 | -.05 | -.06 | -.08 |
| Openness | **.13** | **.22** | .00 | **.09** | **.15** | **.10** | **.11** | **.16** | .01 |
| Truth | **.28** | **.30** | **.14** | **.16** | **.12** | .06 | **.22** | **.13** | **.10** |
| Responsible | **.24** | **.31** | **.15** | .08 | **.13** | .07 | **.12** | **.13** | **.12** |
| Hard-working | **.28** | **.27** | **.16** | **.12** | **.11** | .00 | **.16** | .07 | .05 |
| Forgiving | **.17** | **.28** | **.15** | **.10** | **.16** | **.14** | **.14** | **.12** | **.12** |
| Open-minded | **.21** | **.16** | **.18** | **.17** | **.26** | **.12** | **.22** | **.25** | **.14** |
| Courageous | **.21** | **.26** | **.20** | **.11** | **.19** | **.09** | **.15** | **.18** | **.13** |
| Helpful | **.28** | **.32** | **.16** | **.14** | **.26** | **.11** | **.16** | **.20** | **.17** |
| Loving | **.25** | **.34** | **.18** | **.14** | **.19** | **.11** | **.17** | **.17** | **.17** |
| Capable | **.18** | **.31** | **.15** | **.15** | **.10** | .07 | **.19** | .09 | **.09** |
| Clean | **.16** | **.29** | **.21** | **.13** | **.18** | .03 | **.15** | **.16** | **.10** |
| Self-controlled | **.22** | **.23** | **.19** | **.10** | .07 | .02 | **.13** | .09 | **.08** |
| Independent | **.17** | **.24** | **.12** | **.12** | **.11** | .08 | **.13** | **.11** | **.13** |
| Happy | **.29** | **.29** | **.20** | **.11** | **.15** | .06 | **.14** | **.14** | **.10** |
| Polite | **.19** | **.26** | **.19** | **.09** | **.12** | .05 | **.15** | **.10** | **.13** |
| Intellectual | **.19** | **.24** | **.11** | **.10** | **.17** | **.09** | **.13** | **.14** | .04 |
| Obedient | **.12** | **.18** | **.18** | .00 | .03 | -.01 | .03 | .03 | .05 |
| Logical | **.12** | **.25** | **.10** | **.11** | .07 | .05 | **.08** | .07 | .03 |
| Creative | **.13** | **.28** | **.12** | **.11** | **.25** | **.15** | **.14** | **.25** | **.14** |
| Peace | **.27** | **.33** | **.26** | **.29** | **.30** | **.25** | **.30** | **.29** | **.27** |
| Family Security | **.24** | **.33** | **.18** | .05 | **.08** | .01 | **.13** | **.09** | **.09** |
| Freedom | **.26** | **.26** | **.17** | **.15** | .05 | **.09** | **.21** | **.12** | **.16** |
| Equality | **.23** | **.20** | **.17** | **.25** | **.35** | **.18** | **.29** | **.29** | **.21** |
| Self-respect | **.30** | **.34** | **.22** | **.15** | **.14** | **.11** | **.15** | **.15** | **.17** |
| Happiness | **.32** | **.32** | **.20** | **.10** | **.15** | .07 | **.14** | **.14** | **.16** |
| Wisdom | **.22** | **.34** | **.15** | **.10** | **.18** | **.14** | **.10** | **.17** | **.16** |
| National Security | **.25** | **.27** | **.27** | **.09** | .03 | **.10** | **.13** | .07 | **.20** |
| Salvation | **.15** | **.22** | **.20** | .01 | .04 | .06 | .06 | .06 | **.13** |
| Friendship | **.22** | **.33** | **.20** | **.11** | **.18** | **.10** | **.14** | **.18** | **.16** |
| Accomplishment | **.24** | **.32** | **.17** | **.14** | **.16** | .01 | **.14** | **.13** | .05 |
| Inner Harmony | **.24** | **.35** | **.24** | **.16** | **.25** | **.19** | **.15** | **.25** | **.22** |
| A comfortable life | **.20** | **.24** | **.16** | **.10** | .05 | .02 | **.13** | .07 | **.09** |
| Mature love | **.23** | **.23** | **.20** | **.14** | **.13** | .06 | **.11** | .08 | **.11** |
| Beauty | **.14** | **.25** | **.20** | **.15** | **.31** | **.12** | **.15** | **.30** | **.11** |
| Pleasure | **.17** | **.21** | **.17** | **.13** | **.14** | .04 | **.10** | **.12** | **.13** |
| Recognition | **.17** | **.11** | **.14** | **.12** | **.17** | .03 | .07 | **.10** | .02 |
| Excitement | **.24** | **.17** | **.13** | **.14** | **.19** | .03 | **.15** | **.17** | .06 |
| Leisure satisfaction | **.10** | **.14** | **.14** | .00 | **.13** | .04 | .01 | .06 | .04 |
| Social life satisfaction | **.13** | **.17** | **.13** | .01 | **.15** | .05 | .06 | **.08** | .04 |
| Social Connectedness | **.18** | **.10** | **.09** | .03 | .01 | .03 | .05 | -.05 | -.01 |

*(Continued)*

**Table 5.** (Continued)

| Sample | Health | | | Environment | | | Animal Rights | | |
|---|---|---|---|---|---|---|---|---|---|
| | 1 | 2 | 3 | 1 | 2 | 3 | 1 | 2 | 3 |
| Involved in religious organization | .03 | **.12** | .00 | -.06 | -.05 | -.01 | **-.07** | **-.08** | -.11 |
| Involved in environmental organization | .01 | .06 | -.02 | **.21** | **.25** | **.20** | **.13** | **.17** | **.11** |
| Involved in humanity organization | -.01 | .06 | -.02 | **.13** | **.17** | **.15** | **.12** | **.14** | .04 |
| Visited a museum | -.02 | **.09** | -.03 | **.08** | **.11** | **.13** | .02 | .07 | -.03 |
| Crafts | .02 | **.07** | .04 | **.07** | .07 | **.11** | **.13** | **.08** | **.10** |
| Care for plants/animals | **.07** | **.10** | **.09** | **.12** | **.16** | **.11** | **.17** | **.19** | **.13** |
| Uses Linkedin | **.08** | **.07** | -.07 | -.01 | .07 | -.03 | .00 | .06 | **-.13** |
| Self religious status | **.11** | **.17** | **.08** | -.05 | .00 | .00 | -.02 | -.01 | -.05 |
| Believe in God | **.20** | **.22** | **.15** | -.05 | **-.07** | .05 | .07 | -.01 | .02 |
| Believe in afterlife | **.09** | **.15** | **.09** | -.02 | -.03 | .06 | -.02 | **-.13** | .05 |
| Believe in heaven | **.12** | **.11** | **.09** | -.02 | .01 | .01 | -.04 | **-.14** | .00 |
| Frequency of praying | **.12** | **.16** | **.08** | -.05 | -.02 | .03 | -.03 | -.03 | -.02 |

Significant correlations ($p < .01$) in bold.

Regression models confirmed primary associations between the environment motives and responses to the environment flyer and animal rights motives to the animal rights flyer. The VEMI environment scale emerged as the only significant predictor in the regression model for the health flyer. These preregistered regression models tested associations between vegetarian motives and responses to the flyers, controlling for other vegetarian motives. We conducted exploratory analyses in which we reversed the independent and dependent variables in our regression analyses, to test whether flyers would have specific relations with motives, controlling for the responses to other flyers. In those models, responses to the health flyer emerged as the only significant predictor of the VEMI health scale ($\beta = .15$). Likewise, responses to the environment and animal rights flyers were the only significant predictors of the VEMI environment and animal rights scales, respectively. This pattern indicates that, controlling for general motives to be a vegetarian, there are no specific links between health motives and responses to health-focused advocacy, whereas controlling for general responsivity to advocacy, there may be specific links between health-focused advocacy and health-related vegetarian motives. Overall, the results support the utility of targeting advocacy based on the environment or animal rights to people most likely to care about those issues, and provide weak to mixed support for targeting advocacy based on health motives.

## Discussion

The variety of pathways that can lead a person to vegetarian diet raises the possibility that people who select different pathways are also different in other ways, but little is known about these differences or their importance for eating behavior. Thus, the purposes of this study were to develop a measure of health, environmental, and animal rights motives for vegetarian eating, examine the correlates of these dimensions, and test whether motives differentially predict responses to advocacy materials.

### Vegetarian eating motives inventory

Our first step was to develop the *Vegetarian Eating Motives Inventory* (VEMI), a measure that reliably distinguishes between health, environmental, and animal rights motives for plant-

**Table 6. Variables with significant associations to plant-based eating motives in two convenience samples.**

| Variable | Undergraduate Sample 1 | | | MTurk Sample 2 | | | |
| --- | --- | --- | --- | --- | --- | --- | --- |
| | Best Model | Health | Environment | Animal | Replicate | Health | Environment | Animal |
| vegan | Free All | -.19 | .56 | .19 | Yes | -.17 | .34 | .35 |
| peace | Health Free | .18 | .16 | .15 | Yes | .21 | .15 | .14 |
| agreeableness | Free All | .22 | -.01 | .16 | No | .28 | .10 | .01 |
| truth | Free All | .26 | -.01 | .16 | Yes | .26 | -.01 | .10 |
| responsible | Health Free | .25 | .02 | .02 | Yes | .26 | .02 | .01 |
| hard working | Health Free | .28 | .04 | .03 | Yes | .23 | .02 | .02 |
| forgiving | Health Free | .15 | .05 | .05 | Yes | .19 | .04 | .04 |
| courageous | Health Free | .20 | .05 | .04 | Yes | .19 | .06 | .05 |
| helpful | Health Free | .26 | .05 | .05 | No | .23 | .19 | .00 |
| loving | Health Free | .22 | .06 | .06 | Yes | .23 | .05 | .05 |
| self-controlled | Health Free | .20 | .04 | .04 | Yes | .23 | .02 | .02 |
| independent | Health Free | .16 | .05 | .05 | Yes | .20 | .03 | .03 |
| happy | Health Free | .30 | .03 | .02 | Yes | .26 | .03 | .03 |
| intellectual | Health Free | .17 | .05 | .04 | Yes | .19 | .05 | .04 |
| family security | Free All | .25 | -.08 | .11 | No | .34 | -.04 | .01 |
| freedom | Free All | .24 | .00 | .16 | No | .27 | -.15 | .14 |
| self-respect | Health Free | .28 | .05 | .04 | Yes | .31 | .03 | .03 |
| happiness | Health Free | .32 | .02 | .02 | Yes | .28 | .02 | .02 |
| wisdom | Health Free | .21 | .03 | .03 | Yes | .27 | .03 | .03 |
| national security | Health Free | .23 | .03 | .03 | No | .25 | -.09 | .05 |
| salvation | Health Free | .16 | -.01 | -.01 | Yes | .17 | -.01 | -.01 |
| friendship | Health Free | .21 | .04 | .04 | Yes | .22 | .04 | .04 |
| accomplishment | Health Free | .21 | .05 | .05 | Yes | .24 | .04 | .04 |
| harmony | Health Free | .22 | .06 | .06 | Yes | .26 | .08 | .08 |
| comfort | Health Free | .18 | .04 | .04 | Yes | .20 | .02 | .01 |
| mature love | Health Free | .19 | .05 | .05 | Yes | .19 | .04 | .03 |
| connectedness | Health Free | .18 | -.01 | -.01 | No | .15 | .05 | -.14 |
| environmental organization | Environment Free | -.01 | .22 | -.01 | Yes | -.02 | .28 | -.02 |
| visited opera | Environment Free | -.17 | .16 | -.22 | No | -.01 | .21 | .02 |
| conscientiousness | Environment Free | .10 | -.06 | .15 | Yes | .11 | -.06 | .15 |
| capable | Environment Free | .12 | .01 | .18 | Yes | .14 | -.05 | .20 |
| polite | Environment Free | .12 | -.05 | .17 | Yes | .11 | -.04 | .16 |
| crafts | Animal Free | -.01 | -.02 | .19 | Yes | .03 | -.02 | .15 |
| care for plants/animals | Animal Free | .03 | .04 | .19 | Yes | .04 | .04 | .22 |

Coefficients represent beta weights from Structural Equation Models.

based diets. The scales of this brief instrument were internally consistent and demonstrated a robust factor structure, including measurement invariance across three samples in two languages, men and women, and white and non-white participants. This measure has considerable promise for future research on the motivations for plant-based eating in Western cultures. Moreover, although our goal was to develop the VEMI to assess the potential motives of non-vegetarians in a general population, it can be easily adapted for research among vegans, vegetarians, flexitarians, reducetarians, and other groups. It could also be used at an individual level to better understand the kinds of factors that might be most influential for a particular person. The VEMI thus provides researchers and advocates with a well-validated and flexible

**Table 7. Replicated associations in the LISS sample.**

| Variable | Path Coefficients | | | Pearson Correlations | | |
|---|---|---|---|---|---|---|
| | **Health** | **Environment** | **Animal** | **Health** | **Environment** | **Animal** |
| agreeableness | .15 | .09 | .09 | .21 | .19 | .23 |
| hard working | .21 | -.04 | -.04 | .16 | .00 | .05 |
| courageous | .19 | .02 | .02 | .20 | .09 | .13 |
| loving | .15 | -.01 | .10 | .18 | .11 | .17 |
| self-controlled | .23 | -.09 | .02 | .19 | .02 | .08 |
| happy | .22 | -.01 | -.01 | .20 | .06 | .10 |
| family security | .20 | -.10 | .05 | .18 | .01 | .09 |
| self-respect | .20 | -.02 | .09 | .22 | .11 | .17 |
| happiness | .18 | -.07 | .12 | .20 | .07 | .16 |
| national security | .26 | -.09 | .13 | .27 | .10 | .20 |
| salvation | .19 | -.05 | .08 | .20 | .06 | .13 |
| friendship | .17 | -.03 | .10 | .20 | .10 | .16 |
| accomplishment | .21 | -.07 | .00 | .17 | .01 | .05 |
| harmony | .18 | .08 | .08 | .24 | .19 | .22 |
| comfort | .18 | -.09 | .06 | .16 | .02 | .09 |
| mature love | .22 | -.05 | .04 | .20 | .06 | .11 |
| environmental organization | -.14 | .23 | .05 | -.02 | .20 | .11 |

Coefficients represent beta weights from Structural Equation Models. Variables were included in this table if they replicated results from the first two samples.

measure for assessing the primary motives for plant-based eating among various individuals and groups.

## Eating motivation profiles

We next used the VEMI scale to identify profiles of individuals who are most sympathetic to different reasons to be vegetarian. Overall, findings from three diverse samples suggested that health motives are the most common reason to consider adopting a plant-based diet in general and that health motives have the broadest array of correlates.

A number of criteria reliably correlated with plant-based motives across samples. By this standard, 21 variables correlated with all three motives. The common thread in this list seemed to be a communal orientation to life (e.g., agreeableness, loving, and valuing peace). The profile of people motivated by health was more conventional, as defined by 20 variables (e.g., male, hard-working, obedient, life satisfaction, and religiosity). The only variables that correlated uniquely with environmental motives were openness to experience and having visited a museum. Being involved in a religious organization and doing crafts were uniquely related to

**Table 8. Correlations and regression coefficients indicating associations between VEMI scale scores and attitudes about advocacy flyers.**

| Flyer | Health | | Environment | | Animal Rights | |
|---|---|---|---|---|---|---|
| VEMI Scale | r | β | r | β | r | β |
| Health | .17* | .06 | .09* | -.07 | .15* | -.02 |
| Environment | .32* | .26* | .45* | .39* | .36* | .20* |
| Animal Rights | .25* | .09 | .32* | .14* | .42* | .32* |
| R2 | | .12* | | .21* | | .12* |

\* p < .05

the animal rights motive. Valuing intellectual pursuits was related to both health and environmental motives, whereas being involved in a humanity organization was related to both environmental and animal rights motives. Finally, nine variables were related to both health and animal rights motives. As a group, they seemed to involve morality (e.g., conscientiousness, valuing truth, being self-controlled).

In our primary analytic approach, we used a more restrictive strategy with latent variables to account for measurement error and regression models to identify unique associations with each of the plant-based motives. Based on this approach, people who were primarily motivated by their health tended to be more agreeable, to have instrumental values (i.e. preferred means of achieving goals) involving hard work, courage, love, self-control, being happy, and to have terminal values (i.e., desired end states) involving family security, self-respect, happiness, national security, salvation, friendship, accomplishment, harmony, comfort, and mature love. This pattern paints a picture of a fairly conventional person who views working hard and getting along with others as the formula for a good life. In general, people whose main motives for considering a vegetarian diet are related to their health were not particularly compelled by vegetarian flyers, regardless of their content.

The only criterion uniquely and reliably related to environmental motives was participation in an environmental rights organization. No criteria were reliably related to animal rights motives across all three samples based on our primary analytic strategy. These circumscribed findings for the environment and animal rights scales surprised us given the large number of correlates we examined. This could have to do with our relatively conservative analytic approach, given the larger number of findings based on bivariate correlations that were significant at $p <$ .01. However, by and large these results suggest that few traits, values, hobbies, habits, or demographic characteristics correlate in a way that is both robust and specific to the two major ethical motives for plant-based eating. This may suggest that "ethical vegetarianism" is a moral issue with relative specificity, as exemplified by the large numbers of people who actively promote social justice and environmental protection yet continue to eat animals. While there was some specificity between animal rights/environmental motives and responsivity to animal rights/environmental flyers, a more general finding is that people with ethical motives to consider a vegetarian diet were more responsive to advocacy flyers, including one that emphasized health benefits.

## Implications for targeted advocacy

This pattern of results presents a kind of paradox for targeted advocacy. The most common reason people say they would consider being vegetarian has to do with health, and this study identified factors that could be used to identify those people. However, people driven primarily by health motives are least likely to respond to vegetarian advocacy. One interpretation of these results is that most people care about their health, but most people don't connect health to vegetarian diet because the connection is indeed tenuous empirically. The fact that the most common reason people cite for considering a vegetarian diet is also the least compelling may help explain why there continues to be relatively few vegetarians, and why people motivated by health are also least strict [41,45,61–63] and compliant [1,64,65] with a vegetarian diet. Our data also supports this view somewhat, in that being vegan was more strongly associated with animal and plant motives than health motives in all three samples, although it did not surpass our cutoff in Sample 1 (correlations were .12 with both the animal and environment scales).

Conversely, people who are sympathetic to the ethical arguments for a vegetarian diet cannot easily be distinguished in other ways, but they are most likely to respond to vegetarian advocacy. The one exception is the relatively unsurprising finding that people affiliated with environmental advocacy groups are most likely to respond to an environmental argument

supports the idea of encouraging individuals motivated by such concerns to see the connection between plant-based diets and climate change (e.g., [66]). Indeed, it is likely that many individuals who are passionate about this issue are not fully informed about the negative environmental impact of eating meat [67], and this information gap could be usefully exploited by animal advocacy groups who target individuals with a demonstrated interest in environmental activism.

However, overall these results do not seem to support the utility of selecting advocacy materials based on the kinds of people those materials would target. Instead, these results provide important information about ways in which targeted advocacy might not be productive. For instance, none of the demographic features that are known to be associated with plant-based eating in general, such as being young [44,45], female [1,44,46–49,63] and liberal [45,46,49,52,55–59], were differentially associated with health, environmental, or animal rights motives. The higher rates of vegetarianism among such individuals suggest that they represent fruitful targets for advocacy in general, but the results of this study do not provide guidance about which motives to appeal to among them, in particular.

It is worth noting that approaches to advocacy may depend on the end goal and beliefs about the best way to achieve that goal. Animal rights advocates [29,68] have argued that vegetarian advocacy should always focus on ethical motives. The more practical sector of plant-based diet advocacy (e.g., Leenaert, 2017; Joy, 2008 [30,31]) may be relatively more receptive to emphasizing health as a potential first step in reducing meat consumption. Our results about the specific correlates of health motives may help guide this step. Ultimately, evidence that links motives, advocacy approaches, and behavior change will determine the best way to reduce meat consumption in general, and we suspect that a multipronged approach may prove most effective [69].

## Limitations and future directions

Although we examined a large number of criteria, we were constrained by the data collected by LISS and it is likely that we missed important unmeasured variables that would specifically correlate with different vegetarian motives. Likewise, while health, the environment, and animal rights are the most common motives for plant-based diets in Western societies, certain individuals may have more specific reasons that are not sampled on the VEMI, such as those related to religion or taste. Specificity may also be required to better understand the resistance to vegetarian diets. For instance, concerns have been raised about the difficulties poorer people have in finding healthy plant-based food, and this poses a considerable challenge to plant-based diet advocates for whom positioning one form of social justice (i.e., animal rights) against another (i.e., opportunities for the underprivileged) does little good.

A second major limitation is that the current results do not inform specific strategies to encourage people with different motives to change their diets in practice. For instance, some research suggested that people change their behavior upon becoming more aware of the impacts of eating animals [34,65,70–72], whereas other research suggested that increasing people's awareness alone may not be sufficient to effectively change their behavior [31,73]. This issue sits downstream from the goals of our work, but it is equally critical for the ultimate goals of understanding the transition to vegetarian diets.

Third, in this study we exclusively employed self-report measures because we were interested in consciously accessible motives. However, future work examining attitudes that may be outside of peoples' conscious awareness as well as directly behavioral outcome variables would be a useful extension of the current studies. Fourth, further work could be done to understand the underlying mechanisms of different attitudes towards plant-based dieting and animals

[74]. Fifth, we focused in this study on distinguishing among the three major non-religious motives for vegetarian diet, because research suggests that these are the most common motives in general and because advocacy focuses almost exclusively on these three reasons to avoid meat. However, our results suggest that the VEMI scales could be combined into an overall composite useful for examining motives for vegetarian diet in general, in that the scales were intercorrelated and each distinguished vegan from non-vegan respondents. Moreover, there may be considerable value in assessing motives beyond those measured by the VEMI.

Finally, different approaches to the one taken here may be useful for identifying profiles of people who will tend to respond to different forms of activism. For example, machine learning approaches can be used in very large samples of users to identify an array of online behaviors that may be related to different motives for plant-based diets. This is a powerful tool that may have applicability, for instance in sampling online behavior to produce algorithms that can target specified audiences from within social media platforms [75]. Another is that considering the motives in favor of meat-eating [76] may prove useful in identifying the best way of encouraging plant-based diets. In a previous, preliminary study, we found that health motives were unrelated to motives for eating meat, whereas the environmental and animal rights motives were negatively related to seeing meat eating as "normal" or "nice" [77]. Future work that examines the links between motives to avoid meat and motives to eat meat would accordingly be informative.

## Conclusion

In this study, we developed the Vegetarian Eating Motives Inventory (VEMI), a brief and psychometrically robust measure of the three main motives for adopting a plant-based diet: health, the environment, and animal rights. We used this measure to identify profiles of people most likely to respond to appeals to these different motives and to test whether motives predict responses to advocacy materials. In a general populati0n, health motives are the most common and have the widest array of correlates, which generally involve agentic and communal values. However, people who cite health motives were relatively unresponsive to advocacy materials compared to people who cite environmental or animal rights motives.

## Author Contributions

**Conceptualization:** Christopher J. Hopwood, Wiebke Bleidorn.

**Data curation:** Christopher J. Hopwood, Wiebke Bleidorn, Sophia Chen.

**Formal analysis:** Christopher J. Hopwood, Wiebke Bleidorn, Ted Schwaba, Sophia Chen.

**Funding acquisition:** Christopher J. Hopwood, Wiebke Bleidorn.

**Investigation:** Christopher J. Hopwood, Wiebke Bleidorn.

**Methodology:** Christopher J. Hopwood, Wiebke Bleidorn.

**Project administration:** Christopher J. Hopwood, Wiebke Bleidorn.

**Resources:** Christopher J. Hopwood, Wiebke Bleidorn.

**Software:** Christopher J. Hopwood, Wiebke Bleidorn.

**Supervision:** Christopher J. Hopwood, Wiebke Bleidorn.

**Writing – original draft:** Christopher J. Hopwood.

**Writing – review & editing:** Christopher J. Hopwood, Wiebke Bleidorn, Ted Schwaba.

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
