## [Decision Letter · Decision Letter 0]

12 Feb 2020

PONE-D-19-35075

Health, Environmental, and Animal Rights Motives for Vegetarian Eating

PLOS ONE

Dear Dr. Hopwood,

Thank you for submitting your manuscript to PLOS ONE. After careful consideration, we feel that it has merit but does not fully meet PLOS ONE’s publication criteria as it currently stands. Therefore, we invite you to submit a revised version of the manuscript that addresses the points raised during the review process.

We would appreciate receiving your revised manuscript by Mar 27 2020 11:59PM. To enhance the reproducibility of your results, we recommend that if applicable you deposit your laboratory protocols in protocols.io, where a protocol can be assigned its own identifier (DOI) such that it can be cited independently in the future. For instructions see: http://journals.plos.org/plosone/s/submission-guidelines#loc-laboratory-protocols

We look forward to receiving your revised manuscript.

Kind regards,

Valerio Capraro

Academic Editor

PLOS ONE

Additional Editor Comments (if provided):

I have now received a very detailed review from one expert in the field. The review is largely positive, but suggests a number of minor revisions. Therefore, I would like to invite you to revise your work following these comments. Moreover, your result that love, happiness, and other emotions, are correlated to vegetarianism, made me think about the recent result that priming emotion makes people less anthropocentric speciesist (Caviola & Capraro, Social Psychological and Personality Science, in press). I was wondering whether there is a relationship between these two results. Of course it is not a requirement to cite this in the paper, but I am mentioning because it might be an idea worth thinking about.

Journal Requirements:

Reviewers' comments:

Reviewer's Responses to Questions

**Comments to the Author**

1. Is the manuscript technically sound, and do the data support the conclusions?

Reviewer #1: Yes

2. Has the statistical analysis been performed appropriately and rigorously? 

Reviewer #1: Yes

3. Have the authors made all data underlying the findings in their manuscript fully available?

Reviewer #1: Yes

4. Is the manuscript presented in an intelligible fashion and written in standard English?

Reviewer #1: Yes

5. Review Comments to the Author

Reviewer #1: This is an interesting and well-conceived/executed manuscript with the relevant aim of developing a questionnaire measure (Vegetarian Eating Motives Inventory) to assess the importance attached to (often cited) reasons for choosing a vegetarian or meat reducing diet. The design and the statistics/analysis is in general very high level. Although the paper primarily is a developmental one, there are some interesting results as well. The authors also have a number of points about the ramifications in a general population (where I think they should be a bit more cautious, cf point 10 below).

I have a number of suggestions.

1) You could specify more concretely in the abstract which population the Inventory is for. Only for vegetarians and vegans? Only for meat reducers or the general meat consuming population (which will make the use of the measure more clear for future interested researchers). I liked the paragraph in the Discussion (p 15) which perhaps could be used in a shorter format in the abstract? (“This measure has considerable promise for future research on the motivations for plant-based eating. Moreover, although our goal was to develop the VEMI to assess the potential motives of non-vegetarians in a general population, it can be easily adapted for research among vegans, vegetarians, flexitarians, reducetarians, and other groups.” )

2) Related to point 1: could be pointed out more clearly that, since the inventory exhibits invariance, it is likely to function well throughout Western societies (however, see point 6)

3) It is fine that it is stated that the inventory examines three most dominant reason in Western societies. You could mention already in the Intro that you exclude non-religious motivations that also exists in Western societies (e.g. regligous, hindus).

4) It is know from studies that some vegetarians/vegans develop meat disgust/distaste over time (see for instance your own ref 63). At some point this becomes an important motivation. You should probably mention that this was excluded from the Inventory in the Limitations and Future Directions section. It makes perfectly good sense to me that you excluded it, as you were interested in non-vegetarian converters (that arguably have not developed disgust).

5) For all samples, it should described whether there were exclusion criteria in general and owning to diet specifically (could any type of diet (meat eating status) participate?). The geographical and/or national composition of all samples should be specified directly in the Methods section (can only see that Sample 3 is Dutch). Also, considering the nature of your research endeavor, it would also be a good idea to report in the Method section how many vegetarians, vegans, meat reducers there were in each sample.

6) Your claim of invariance could be strengthened and improved if further multigroup measurement invariance assessments were carried out. You could look at gender (in all samples), age (in sample 2 and 3 where there is much age variation). For sample 1 you could also determine “broader cultural” invariance. Thus, you have students with nonwestern heritage (sample 1: 485 Asians). The analysis may not be strictly necessary, but it seems just as important to look at invariance related to actual empirical differences and not only sample differences.

7) I like the introduction and motivation for developing the inventory very much – it is clear and to the point. Many relevant former measures are discussed.

On page 3: you write “ a more ethical approach to inter-species relationships”. What is meant with a more ethical approach? This appears too normatively motivated. An anthropocentric ethical approach is just as ethical as an animal rights approach. Just another kind of ethics. Please reformulate to the specific ethics approaches (animal rights or whatever) that the references suggest.

You could say that you also collected a sample 4 in the latter paragraph before the method section (p7), since you also reported about the three other samples in the Intro.

8) Tables:

Table 5: please add notes to the tables specifying a) what the numbers in the table indicate (betas?), b) for each row whether it was just one or all of the scales that had to be significant for the association to be reported in the table, and 3) type of analysis (distributional-wise (logit, linear or whatever)) the table reports from.

Table 6: please add notes to the tables specifying a) what type of correlational test that is reported (pearson, spearman, etc).

The table 7 note: “association predicted in LISS” should be phrased to “prediction of associations to be tested in LISS” to avoid confusion.

9) In the limitation section (p 20) you mention that a useful extension could be to study motivations outside of peoples’ conscious awareness. I am not sure they are motivations anymore when they move away from awareness, but maybe implicit attitudes. This has been studied for meat eaters actually. Anyway, another way to look at it, and that you might consider including as a future research avenue, could be to prompt people for what is most important (best-worse scaling)

10) The design of the correlational analyses:

My primary concern of the study relates to the methodology of the correlational analyses. You

i) use two convenience samples (from different populations) to determine the Correlates that are looked for in the Plant-Based Eating Motives in Sample 3 (LISS).

ii) Following exclusions on basis of i) and what could be replicated in the LISS sample, you conclude quite much about relevant criterions (p17 “The only criterion reliably related to environmental motives was participation in an environmental rights organization. No criteria were reliably related to animal rights motives across all three samples. These circumscribed findings for the environment and animal rights scales surprised us given the large number of correlates we examined” and later that (p 21) “health motives have the widest array of correlates, which generally involve agentic and communal values.”).

This makes me wonder whether you may be relying too much on the two convenience samples to exclude possible relevant findings (and ACTUAL population differences) from sample 3?

In general, I think that the idea about determining “relevant/valid” criterion variables through three samples need more clarification. There are a number of questions that I am left with:

A) First of all, why did you choose to do three samples to determine the final (potential) relevant criterions? Why not five? Or ten?

B) Is is valid to study this with varying populations (students, Americans (I suppose, in sample 2), and people from the Netherlands). Any of the correlations (or non-assiciations) that could not be replicated in sample 2 and 3 could be a consequence of actual associational differences in the populations compared to the baseline in sample 1 (sample 1 consists of soon to be highly educated and young individuals – and many women that also could have a big impact).

In short, I do not think that the preregistration procedure related to the Sample 3 analysis is particular useful considering the aim of detecting relevant associations in a general population because of the aforementioned points A and B. I realize that you have put much effort into this part of the analysis and I am not asking you to remove this (you may also disagree with my remarks). But you could at least consider mentioning some of these limitations in your discussion (and rebut them if you disagree). But your claims about the general population (returning to point ii above) also rest on this procedure and you should as a minimum state in the Conclusion that the population results (your array of correlates) may be incomplete because of the measure developmental character of the paper and that future studies should examine this further.

11) Following up on the former: you mention “large effects distinguished the 97 vegans across all three samples from non-vegan respondents for the health (d = .51), environment (d = 1.29), and animal rights (d = .97). But you report no significant correlations with vegans in sample 3 - LISS data Table 7. Because there was a little correlation in sample 1 (r 0.12). Could this be because of too few vegans in sample 1 so the small correlation is by chance? Related to this: I cannot understand that you mention that you excluded correlational candidate variables in sample 1 and 2 if “less than 50 participants responding either “no” or “yes” – p 13.) But if you had an inclusion criteria of 50 in the two first samples and there only are in total 97 vegans in all three samples I cannot see how vegans should be included in the pool of correlational variable in the first place.

Could you please explain (I may be overseeing something)?

12) It seems that you excluded “All zero” from the subsequent follow-up correlation identification. Why? It seems just as important to show that non-associations replicate (this is a case of discriminant validity). Please explain why you were conceptually interested only in replications of significant associations.

6. PLOS authors have the option to publish the peer review history of their article (what does this mean?). If published, this will include your full peer review and any attached files.

Reviewer #1: No

---

## [Author Response · Author response to Decision Letter 0]

27 Feb 2020

Editor

1. Your result that love, happiness, and other emotions, are correlated to vegetarianism, made me think about the recent result that priming emotion makes people less anthropocentric speciesist (Caviola & Capraro, Social Psychological and Personality Science, in press). I was wondering whether there is a relationship between these two results. Of course it is not a requirement to cite this in the paper, but I am mentioning because it might be an idea worth thinking about.

Thank you for mentioning this paper which we read with interest. We had a difficult time connecting the two findings directly, though. Caviola and Capraro found that priming emotion makes people less speciesist – a really interesting finding. We found that love and some other values (as opposed to emotions) were related to the health motive to consider being vegetarian. In some sense, this motive is the least “vegetarian” in that it does not seem to be driven by ethical concerns, and overall the pattern of correlates makes it seem like the most conventional of the three motives. Basically, most people want to be healthy whereas fewer people care about the environment and even fewer about animal rights, so health motives correlate with a lot of other fairly normative variables. However, we do think that it would be interesting to do experimental work with vegetarian motives, and in particular to see how affect intersects with cognitive and motivational dynamics. Thus, we have included this reference and the value of future experimental work in this area as a future direction in the Discussion. 

Reviewer

This is an interesting and well-conceived/executed manuscript with the relevant aim of developing a questionnaire measure (Vegetarian Eating Motives Inventory) to assess the importance attached to (often cited) reasons for choosing a vegetarian or meat reducing diet. The design and the statistics/analysis is in general very high level. Although the paper primarily is a developmental one, there are some interesting results as well. The authors also have a number of points about the ramifications in a general population (where I think they should be a bit more cautious, cf point 10 below). I have a number of suggestions.

Thank you for your positive comments and your very helpful suggestions below. 

1) You could specify more concretely in the abstract which population the Inventory is for. Only for vegetarians and vegans? Only for meat reducers or the general meat consuming population (which will make the use of the measure more clear for future interested researchers). I liked the paragraph in the Discussion (p 15) which perhaps could be used in a shorter format in the abstract? (“This measure has considerable promise for future research on the motivations for plant-based eating. Moreover, although our goal was to develop the VEMI to assess the potential motives of non-vegetarians in a general population, it can be easily adapted for research among vegans, vegetarians, flexitarians, reducetarians, and other groups.” )

This is a good idea. We have added this to the abstract. 

2) Related to point 1: could be pointed out more clearly that, since the inventory exhibits invariance, it is likely to function well throughout Western societies (however, see point 6)

We now mention this in the Discussion. 

3) It is fine that it is stated that the inventory examines three most dominant reason in Western societies. You could mention already in the Intro that you exclude non-religious motivations that also exists in Western societies (e.g. regligous, hindus).

We now mention this in the Introduction. 

4) It is know from studies that some vegetarians/vegans develop meat disgust/distaste over time (see for instance your own ref 63). At some point this becomes an important motivation. You should probably mention that this was excluded from the Inventory in the Limitations and Future Directions section. It makes perfectly good sense to me that you excluded it, as you were interested in non-vegetarian converters (that arguably have not developed disgust).

We now mention taste as an alternative motive in the Discussion section. 

5) For all samples, it should described whether there were exclusion criteria in general and owning to diet specifically (could any type of diet (meat eating status) participate?). The geographical and/or national composition of all samples should be specified directly in the Methods section (can only see that Sample 3 is Dutch). Also, considering the nature of your research endeavor, it would also be a good idea to report in the Method section how many vegetarians, vegans, meat reducers there were in each sample.

We now provide this information in the Methods section. 

6) Your claim of invariance could be strengthened and improved if further multigroup measurement invariance assessments were carried out. You could look at gender (in all samples), age (in sample 2 and 3 where there is much age variation). For sample 1 you could also determine “broader cultural” invariance. Thus, you have students with nonwestern heritage (sample 1: 485 Asians). The analysis may not be strictly necessary, but it seems just as important to look at invariance related to actual empirical differences and not only sample differences.

We now test invariance across men and women in the first three samples as well as white vs. non-white in the first two (predominately North American) samples. 

7) I like the introduction and motivation for developing the inventory very much – it is clear and to the point. Many relevant former measures are discussed. 

Thank you for this positive comment.

8) On page 3: you write “a more ethical approach to inter-species relationships”. What is meant with a more ethical approach? This appears too normatively motivated. An anthropocentric ethical approach is just as ethical as an animal rights approach. Just another kind of ethics. Please reformulate to the specific ethics approaches (animal rights or whatever) that the references suggest.

This is a good point – we have changed this word to “humane”. 

9) You could say that you also collected a sample 4 in the latter paragraph before the method section (p7), since you also reported about the three other samples in the Intro.

We now state this in the Introduction. 

8) Tables:

Table 5: please add notes to the tables specifying a) what the numbers in the table indicate (betas?), b) for each row whether it was just one or all of the scales that had to be significant for the association to be reported in the table, and 3) type of analysis (distributional-wise (logit, linear or whatever)) the table reports from.

Table 6: please add notes to the tables specifying a) what type of correlational test that is reported (pearson, spearman, etc).

The table 7 note: “association predicted in LISS” should be phrased to “prediction of associations to be tested in LISS” to avoid confusion.

We have made each of these corrections to the Table notes, titles, and/or Methods/Results sections. 

9) In the limitation section (p 20) you mention that a useful extension could be to study motivations outside of peoples’ conscious awareness. I am not sure they are motivations anymore when they move away from awareness, but maybe implicit attitudes. This has been studied for meat eaters actually. Anyway, another way to look at it, and that you might consider including as a future research avenue, could be to prompt people for what is most important (best-worse scaling)

We have modified our language around this issue in the Limitations section. 

10) The design of the correlational analyses:

My primary concern of the study relates to the methodology of the correlational analyses. You

i) use two convenience samples (from different populations) to determine the Correlates that are looked for in the Plant-Based Eating Motives in Sample 3 (LISS).

ii) Following exclusions on basis of i) and what could be replicated in the LISS sample, you conclude quite much about relevant criterions (p17 “The only criterion reliably related to environmental motives was participation in an environmental rights organization. No criteria were reliably related to animal rights motives across all three samples. These circumscribed findings for the environment and animal rights scales surprised us given the large number of correlates we examined” and later that (p 21) “health motives have the widest array of correlates, which generally involve agentic and communal values.”).

This makes me wonder whether you may be relying too much on the two convenience samples to exclude possible relevant findings (and ACTUAL population differences) from sample 3?

In general, I think that the idea about determining “relevant/valid” criterion variables through three samples need more clarification. There are a number of questions that I am left with:

A) First of all, why did you choose to do three samples to determine the final (potential) relevant criterions? Why not five? Or ten?

B) Is is valid to study this with varying populations (students, Americans (I suppose, in sample 2), and people from the Netherlands). Any of the correlations (or non-assiciations) that could not be replicated in sample 2 and 3 could be a consequence of actual associational differences in the populations compared to the baseline in sample 1 (sample 1 consists of soon to be highly educated and young individuals – and many women that also could have a big impact).

In short, I do not think that the preregistration procedure related to the Sample 3 analysis is particular useful considering the aim of detecting relevant associations in a general population because of the aforementioned points A and B. I realize that you have put much effort into this part of the analysis and I am not asking you to remove this (you may also disagree with my remarks). But you could at least consider mentioning some of these limitations in your discussion (and rebut them if you disagree). But your claims about the general population (returning to point ii above) also rest on this procedure and you should as a minimum state in the Conclusion that the population results (your array of correlates) may be incomplete because of the measure developmental character of the paper and that future studies should examine this further.

This is a good point, and one that we struggled with during our preregistration and in interpreting the results and preparing the manuscript. We agree with the reviewer’s general concern that our approach was perhaps too conservative. We of course want to leave the preregistered results in the paper, but we have now supplemented them with a much less conservative approach, in which we simply retain results that have replicated as bivariate correlations (as opposed to regression weights) across samples. This leads to a much larger number of significant findings. Of course, this might be critiqued as too liberal, but now the reader can choose between a liberal or a conservative approach, or perhaps something between. Given that we preregistered one approach, we continue to lean most heavily on those results in interpreting study findings. 

11) Following up on the former: you mention “large effects distinguished the 97 vegans across all three samples from non-vegan respondents for the health (d = .51), environment (d = 1.29), and animal rights (d = .97). But you report no significant correlations with vegans in sample 3 - LISS data Table 7. Because there was a little correlation in sample 1 (r 0.12). Could this be because of too few vegans in sample 1 so the small correlation is by chance? Related to this: I cannot understand that you mention that you excluded correlational candidate variables in sample 1 and 2 if “less than 50 participants responding either “no” or “yes” – p 13.) But if you had an inclusion criteria of 50 in the two first samples and there only are in total 97 vegans in all three samples I cannot see how vegans should be included in the pool of correlational variable in the first place. Could you please explain (I may be overseeing something)?

We did not compute correlations for these variables in samples 1 and 2 because fewer than 50 people endorsed being vegan. That is why, across all three samples, there were only 97 vegans total. We did not have enough power to test moderation between vegan and non-vegan respondents; we are currently collecting data for that purpose. We included all participants in the absence of evidence for moderation and given that our goal was to assess motives for people in general, regardless of their dietary preferences. 

12) It seems that you excluded “All zero” from the subsequent follow-up correlation identification. Why? It seems just as important to show that non-associations replicate (this is a case of discriminant validity). Please explain why you were conceptually interested only in replications of significant associations.

Our strategy was to interpret only those findings that replicated across the first three samples as significant, and all others as not significant (even if they might have been for one or two samples). While we appreciate the reviewer’s point, we think that this makes for a cleaner and more focused presentation, with the comfort that data and scripts are available to any researcher who wished to do this kind of test relatively easily.

---

## [Editor Report · Decision Letter 1]

5 Mar 2020

Health, Environmental, and Animal Rights Motives for Vegetarian Eating

PONE-D-19-35075R1

Dear Dr. Hopwood,

We are pleased to inform you that your manuscript has been judged scientifically suitable for publication and will be formally accepted for publication once it complies with all outstanding technical requirements.

With kind regards,

Valerio Capraro

Academic Editor

PLOS ONE
---

## [Editor Report · Acceptance letter]

11 Mar 2020

PONE-D-19-35075R1 

Health, Environmental, and Animal Rights Motives for Vegetarian Eating 

Dear Dr. Hopwood:

I am pleased to inform you that your manuscript has been deemed suitable for publication in PLOS ONE. Congratulations! Your manuscript is now with our production department. 

With kind regards,

on behalf of

Dr. Valerio Capraro 

Academic Editor

PLOS ONE